# An update on oral clinical courses among patients with severe acute respiratory syndrome coronavirus 2 (SARS-CoV-2) infection: A clinical follow-up (a prospective prevalent cohort) study

Heron Gezahegn Gebretsadik 🄸 *

School of Global Health & Bioethics, Euclid University, Banjul, Gambia

* gezahegn.heron@gmail.com

## Abstract

### Introduction

Contemporary literature has revealed that Severe Acute Respiratory Syndrome Coronavirus 2 (SARS-CoV-2) causes acute sialadenitis and related symptoms, such as discomfort, pain, swelling, and secretory dysfunction in salivary glands. The secretory dysfunction is due to SARS-CoV-2 infection-induced xerostomia and other associated clinical courses such as sore tongue, mucosal ulcer, and gingivitis in the oral cavity. Furthermore, it has been reported that COVID-19 causes the development of other oral manifestations.

### Materials and methods

A prospective clinical follow-up (a prevalent cohort) study was conducted to identify the possible oral manifestations of SARS-CoV-2 infection among patients admitted toat the Eka General Hospital COVID-19 treatment center. Furthermore, the study aimed to calculate the prevalence rate of oral clinical courses in the cohorts. The study consisted of two follow-up phases: Hospital and patient-home-based.

### Results

A total of 55 patients (36 males and 19 females) met the inclusion criteria and were followed for 7.5 weeks. The 3.5 weeks hospital-based prospective follow-up study documented an 18% (n = 10) prevalence rate of oral clinical courses among the cohorts. Twelve oral symptoms appeared in these ten patients. The manifested oral symptoms were oral mucosal lesions (n = 6), xerostomia (n = 5), and thickening of saliva (n = 1). The oral mucosal lesions per se consisted of aphthous lesions (n = 3), candidiasis (n = 1), geographic tongue (n = 1), and localized gingivitis (n = 1). The four weeks' home-based follow-up study disclosed four newly manifested oral symptoms: hemorrhagic crust, bulla, buccal mucositis, and petechiae. These manifestations appeared among six patients (four males and two females) who had not manifested any oral symptoms during the hospital-based follow-up.

**Data Availability Statement:** All relevant data are within the paper and its Supporting Information files.

**Funding:** The author(s) received no specific funding for this work.

**Competing interests:** The authors have declared that no competing interests exist.

Accordingly, the overall prevalence of oral clinical courses among patients presented with SARS-CoV-2 is raised from 18% (n = 10) to 29% (n = 16). Similarly, the number of clinical courses increased from 12 to 16 after four additional weeks of follow-up.

## Discussion

The study's findings suggest the importance of initiating oral health care for patients with COVID-19. Therefore, multidisciplinary healthcare approaches should be delivered to assure optimal health outcomes. Accordingly, oral health professionals must be a substantial part of the interdisciplinary approach in caring for patients with COVID-19.

## Introduction

Outbreaks of newly emerged respiratory infections are a continuing threat. Over the last two centuries, influenza pandemics occurred about once every thirty years [1]. There is also a continuing threat of new respiratory infections, as illustrated by the emergence of the severe acute respiratory syndrome (SARS-CoV) in 2003 and its potential re-emergence [2]. Outbreaks of diseases might occur over a vast area (several countries or continents) and usually affect a large proportion of the population. The Spanish Influenza pandemic in 1918 is a good example [3]. The 2009 H1N1 influenza virus (previously called "novel H1N1" or "swine flu") which emerged after 90 years in the spring of 2009 is the most recent influenza pandemic in history [4]. The emergence of the Middle East Respiratory Syndrome Coronavirus (MERS-CoV) in 2012 can also be regarded as a major outbreak in this period [5].

In February 2019, the World Health Organization announced the official name of the illness caused by the new coronavirus as Covid-19; the virus is named SARS-CoV-2 [6, 7]. Although the new COVID-19 virus differs from SARS-CoV, it uses the same host receptor, human angiotensin-converting enzyme 2 (ACE2) [8]. Following the rapid expansion of the disease globally, the world health organization declared the condition a pandemic on Mar 11, 2020 [6].

Transmission of viruses can generally occur in two ways: either through relatively large droplets of respiratory fluid (10–100 μm) or through smaller particles called aerosols (<10 μm) [1, 9]. The larger droplets are quickly pulled to the ground by gravity, therefore transmission requires close physical proximity. In contrast, the aerosolized transmission may occur over more considerable distances and does not necessarily require infected and susceptible individuals to be co-located simultaneously [10]. Respiratory and salivary droplets appear to be the main transmission routes of COVID-19 disease through inhalation, ingestion, and/or direct mucous contact [11]. Indeed, it has been suggested that such droplets can travel up to four meters with an uncovered cough [12]. It has also been shown that the SARS-CoV-2 virus can survive in aerosols in an experimental setting. Still, it is unclear to what extent such particles are generated in real-life situations and whether such particles are sufficient to cause an infection [13]. Therefore, the aerosol route for COVID-19 transmission requires further verification in clinical settings, considering the presence of patients and health workers, air circulation, and other environmental factors [14]. The preventive disease approaches are designed to interrupt the SARS-CoV-2 virus transmission [11]. Accordingly, getting a vaccine, wearing a face mask, social distancing, avoiding crowded spaces, proper handwashing, and using disinfectants are the main preventive measures to contain the disease transmission [15]. Patients with COVID-19 are reported to manifest several clinical symptoms. The characteristic

symptoms of the disease are similar to the common flu: fever, dry cough, shortness of breath, and tiredness—it's not solely a running nose [16]. Diarrhea may also manifest in some patients. These symptoms are usually mild [1]. Older persons and people with pre-existing medical conditions (comorbidities) appear to develop a severe illness more often than others [17]. In more severe cases, an infection can cause pneumonia, severe acute respiratory syndrome, kidney failure, and death [10].

Additionally, most viral infections are reported to present a variety of oral manifestations [18]. Oral presentations of viral infections may be a preliminary sign of disease, a vital co-symptom of viral disease, or the only sign observed in such viral illness [15, 19]. Viral infections are the most common reason for oral ulcerations and blisters [20]. Blisters, ulcers, color variation, punctuated hemorrhages, bullae, erythematous plaques, surface/textural changes, and Kaposi's sarcoma are the most typical clinical manifestations associated with different viral diseases such as HIV/AIDS, Chickenpox, and Zika [15]. Hence, the expectation of potential oral clinical manifestations in patients with COVID-19 is warranted [19].

Moreover, a shred of contemporary literature has revealed the development of oral clinical courses in patients with SARS-CoV-2 infection [21]. In mild cases, oral mucosal lesions developed before or at the same time as the initial respiratory symptoms; however, in those who required medication and hospitalization, the lesions developed approximately seven to 24 days after onset symptoms [8, 15]. Oral mucosal lesions are more likely to present as coinfections and secondary manifestations with multiple clinical aspects [15]. Furthermore, various cutaneous manifestations, including varicelliform lesions, pseudochilblain, erythema multiforme (EM)-like lesions, urticaria form, maculopapular, petechiae and purpura, mottling, and livedo reticularis-like lesions are reported in patients with COVID-19 [22]. Despite the probable relationship between the oral cavity and SARS-CoV-2, to date, many variables could influence the presence of the oral manifestations [17]. Most patients take a large number of drugs that may cause oral presentations. Thus the need to evaluate COVID-19-induced oral manifestation of COVID19 in hospitalized patients should be underscored [23].

On the other hand, the pandemic caused significant changes in the current approach to several practices globally [24]. Several initiations have been enhanced to limit the spread of the virus throughout the world [25]. The application of a multidisciplinary approach to tackle the pandemic successfully is one of them [26]. Accordingly, various health professionals from different disciplines have begun to follow a similar approach to treating patients with COVID-19 [2]. This study aimed to suggest the importance of involving dental practitioners in the healthcare team to treat patients with SARS-CoV-2 infection.

## Materials and methods

### Study design

A prospective clinical follow-up (a prevalent cohort) study was conducted to identify potential oral clinical courses as study variables among patients presented with SARS-CoV-2 infection. In addition to identifying these study variables, the study aimed to calculate the prevalence rate of oral clinical courses in the cohorts. Initially, patients admitted to the Eka General Hospital were followed prospectively for an average of 3.5 weeks. Following the discharge of the patients, the follow-up continued for an additional four weeks at their homes.

### Study location and population

The first follow-up study was conducted at the Eka General Hospital COVID-19 treatment center, Addis Ababa, Ethiopia. The second follow-up was carried out in an outpatient setting at the patients' homes. Fifty-five (36 men and 19 women) responsive, volunteer and newly

admitted patients to the hospital between 19 and 25 October 2020 were involved in both follow-up studies. The follow-up studies were comprised of patients from all age groups and lasted till 15 January 2021. The average age of the men (15–80 years) and women (8–90 years) study participants were 53.3 and 50.3 years, respectively. The age of the overall cohorts ranged from 8 to 90 years.

**Inclusion criteria.**

- Responsive and volunteer (parents or guardian consent for minors younger than 18 years old) patients of all age groups with confirmed SARS-CoV-2 infection

- Study participants willing to be involved in both follow-up studies

**Exclusion criteria.**

- Confirmed cases of SARS-CoV-2 infection with oral clinical courses before admission

- Confirmed cases with SARS-CoV-2 infection and had a recent history of viral diseases such as HIV/AIDS, Hepatitis, Chickenpox, etc

- All confirmed patients with SARS-CoV-2 infection that were not responsive or willing to voluntarily participate in the study

- All intubated patients (to avoid associated iatrogenic lesions)

- Patients taking antibiotics

- Patients with systemic diseases such as diabetic Mellitus and other patients with known immune suppression conditions

- Patients residing out of Addis Ababa

## Data collection and analysis

Following the ethical approval by the responsible authorities and the Hospital Research Committee, the relevant primary data was collected by calibrated dentists using the modified World Health Organization (WHO) oral health assessment forms (MOAF) (S1 and S2 Figs). All potential COVID-19-induced oral clinical courses that may manifest during the follow-up period were the study variables. The characteristics of the most common oral lesions were described and defined clearly in the MOAF for the examining clinicians (calibrated dentists) to avoid potential diagnosis bias. The data collectors used the required personal protective instrument (PPE), wooden spatula, and headlight to examine the cohort's oral cavity. Patients were examined twice a week in the hospital setting and at their homes after discharge. Accordingly, the clinical findings in the oral cavity were documented in the MOAF by the examining clinicians.

Eventually, the completed MOAF was collected, cleaned, interpreted, and compiled into an excel sheet. The data collection was followed by data processing, data cleaning, and then data analysis using STATA software. Accordingly, the excel data was imported to STATA. The dataset was then saved in STATA format. The STATA categorized the entered dataset by variables such as gender, age, and oral clinical courses. After creating a "do" file in the STATA, the researcher executed the STATA commands for the data analysis. Finally, the data were analyzed, described, and communicated around the core objectives of the study.

## Ethical clearance

Ethical clearance was obtained from the Addis Ababa Public Health Research and Emergency Management Directorate. The approval number of the clearance statement is "A/A/11247/

227". The Eka General Hospital Ethics Committee also approved the research. Additionally, the study participants freely gave their oral consent to participate in the study. Parents' or guardians' oral consent was obtained for minors younger than 18 years old. There was no form of coercion. Thus, the participants were free from any pressure related to the research.

Furthermore, the researchers informed the participants about the research's purpose and potential benefits. The brief sessions emphasized that participation was voluntary and that the participants had the right to withdraw participation at any time without prejudice. Medical information was kept confidential. Additionally, the researcher used patients' initials in the MCRFs and excel sheet for anonymity.

## Results

This cohort study had two phases of follow-up. The first follow-up phase was carried out in the hospital setting for an average of 3.5 weeks. The second follow-up phase was home-based and took four weeks. The cohorts were the same in both stages. The home-based follow-up aimed to update the previously published hospital-based findings. The findings of the two follow-ups and the overall study are presented below chronologically.

### Findings pertaining to the hospital-based setting

The hospital-based study had a prevalence rate of 18% (n = 10). A total of 12 oral symptoms appeared in these ten patients. More males (n = 7) than females (n = 3) have presented the symptoms. The symptoms were oral mucosal lesions (n = 6), xerostomia (n = 5), and thickening of saliva (n = 1). The oral mucosal lesions include aphthous lesions (n = 3), candidiasis (n = 1), geographic tongue (n = 1), and localized gingivitis (n = 1). Two of the aphthous lesions were developed on the buccal mucosa. The other aphthous lesion was seen on the dorsal midline of the tongue. The tongue (n = 3), buccal mucosa (n = 2), and upper anterior gingiva (n = 1) were the most common sites of presentation of the oral mucosal lesions in this follow-up study (Table 1). Out of the ten cohorts who presented oral symptoms, 2 exhibited two oral manifestations each. In comparison, the remaining eight cohorts had shown only one oral symptom each.

### Findings pertaining to the home-based setting

The home-based follow-up disclosed four newly manifested oral symptoms. The manifestations include hemorrhagic crust (n = 2), bulla (n = 1), buccal mucositis (n = 2) and petechiae (n = 1). These manifestations appeared in 6 patients (four males and two females) who had not manifested any oral symptoms during the hospital-based follow-up. Accordingly, the home-based follow-up yielded a prevalence rate of 11% (n = 6). Patients who had developed oral

**Table 1. Oral clinical courses (√) manifested among patients with SARS-CoV-2 infection during the hospital-based follow-up.**

| Cohorts with oral clinical courses | Oral clinical courses manifested (anatomic landmark involved) | | | | | | Total clinical courses manifested |
|---|---|---|---|---|---|---|---|
| | Xerostomia | Thickening of saliva | Oral mucosal lesions | | | | |
| | | | Aphthous lesion | Candidiasis | Geographic tongue | Erythematous and oedematous gingiva | |
| Females (n = 3) | √ √ √ | √ | | | | | n = 4 |
| Males (n = 7) | √ √ | | 1 (tongue) 2 (buccal mucosa) | 1 (tongue) | 1 (tongue) | 1 (Gingiva) | n = 8 |
| Total (n = 10) | n = 5 | n = 1 | n = 3 | n = 1 | n = 1 | n = 1 | n = 12 |
| | | | n = 6 | | | | |

**Table 2. Oral clinical courses manifested among patients with SARS-CoV-2 infection during the home-based follow-up.**

| Cohorts with oral clinical courses | Oral clinical courses manifested (anatomic landmark involved) | | | | Total clinical courses manifested |
|---|---|---|---|---|---|
| | Hemorrhagic crust | Bulla | Buccal mucositis | Petechiae | |
| Females (n = 2) | 1 (corner of the mouth) | 1 (tongue) | - | - | 2 |
| Males (n = 4) | 1 (lower lip) | - | 2 (buccal mucosa) | 1 (palatal) 1 (lower lip) | 5 |
| Total (6) | 2 | 1 | 2 | 2 | 7 |

symptoms during the hospital-based follow-up were further included in the home-based follow-up study. However, none of them manifested additional oral symptoms.

Furthermore, out of the six cohorts who presented oral symptoms during the home-based follow-up, one male exhibited two oral manifestations (palatal petechiae and lower lip hemorrhagic crust). In contrast, the remaining five cohorts had shown only one oral symptom each. In this follow-up study, lower lip and buccal mucosa were the most frequently (n = 2) involved anatomical landmarks. The other additional sites that present the oral symptoms are the tongue, corner of the mouth, and palate (Table 2).

## Updated findings

A total of 55 patients (36 males and 19 females) fulfilling the inclusion criteria were included in the cohort. The age of the cohorts ranged from 8 to 90 years. However, 78.2% of the cohorts were older than 40 years. The cohorts were followed for an average of 3.5 and 4 weeks in the hospital and home settings, respectively. The overall prevalence of oral clinical courses after both follow-up studies was 29% (n = 16). The oral clinical courses include xerostomia (n = 5), thickening of saliva (n = 1), aphthous lesions (n = 3), candidiasis (n = 1), geographic tongue (n = 1), localized gingivitis (n = 1), hemorrhagic crust (n = 2), bulla (n = 1), petechiae (n = 2), and buccal mucositis (2). Xerostomia (n = 5) and aphthous lesions (n = 3) were the most common oral manifestations among patients with COVID-19. At the same time, the most common anatomical landmarks involved were the tongue (n = 4) and buccal mucosa (n = 4). The oral clinical courses were exhibited in 5 females and 11 males. The females to male ratio (5:11) who had manifested oral clinical courses were comparable to the overall gender proportion of the cohort (34.5% females and 65.5% males). The proportion of cohorts (25%) who had presented oral clinical courses under the age of 40 was also comparable to their overall contribution (21.8%) to the total cohorts. Out of the total 16 cohorts who had presented oral symptoms, three had exhibited two oral manifestations each. In comparison, the remaining 13 cohorts had shown only one oral symptom each.

## Discussion

Scholars have identified oral clinical courses in patients with HIV, Hepatitis, and Zika virus infections [27]. Candidiasis, periodontitis, salivary gland disease, sarcoma, Kaposi's sarcoma, oral hairy leukoplakia, and aphthous ulcers are some of the oral clinical manifestations of HIV infection [28, 29]. Oral lesions such as petechiae and ulcers have been reported but scarcely described among patients with Zika virus infection [30]. Similarly, oral mucosal manifestations have occasionally been described in published literature for dengue cases, including gingival ulcers and petechiae and blisters at the junction of the hard and soft palate [31]. Furthermore, hepatitis virus infections have also demonstrated significant oral manifestations such as gingivitis and gingival bleeding, pseudo-pocket formation, and a mucosal ulcer [32]. On the other hand, several pieces of literature have outlined ACE2 as an essential member of the renin-angiotensin system [33]. ACE2 is widely distributed in the vasculature and participates in

regulating blood pressure [34]. ACE2 protein can also be found in other organs, such as the small intestine, testes, adipose tissue, thyroid gland, kidneys, heart muscle, colon, ovaries, and salivary glands [17, 35]. Accordingly, salivary glands could be the invasive target of SARS-CoV-2 [36]. Moreover, the salivary gland cells with ACE2 receptors may become host cells for the virus and cause inflammatory reactions in related organs and tissues, such as the tongue, the periodontal tissues, and oral mucosa [33, 37]. Furthermore, scholars have revealed oral clinical courses such as dysgeusia, petechiae, gingivitis, candidiasis, traumatic ulcers, geographical tongue, and thrush-like ulcers among patients with SARS-CoV-2 infection [38, 39]. Other contemporary literature has ruled out acute sialidases, which cause salivary gland dysfunction and related discomfort, pain, and swelling [40]. On the other hand, chronic sialadenitis causes xerostomia, sore tongue, mucosal ulcer, and gingivitis in patients with COVID-19 [41]. Xerostomia has been found mainly among COVID-19 patients due to the neuroinvasive and neurotropic potential of SARS-CoV-2 [31]. Additionally, a number of observational studies across the world have reported several oral clinical courses in patients with COVID-19 [15, 42, 43]. Chinese researchers have identified xerostomia in a relatively high proportion of patients [44]. Similarly, a cross-sectional survey of 108 patients with confirmed SARS-CoV-2 in China observed that 46% reported dry mouth, among other symptoms. A human observational study of 20 patients has revealed nearly 30% of xerostomia during hospitalization [45]. Other related research works have reported that dry mouth, dysgeusia, oral ulcerations, and opportunistic infections are the most common oral manifestations expressed in COVID-19-positive patients [46, 47]. Similarly, the findings of this prevalent cohort study have revealed a considerable report of xerostomia. The prevalence of xerostomia was 9% (n = 5) among the cohorts involved in this follow-up study. Another literature review found aphthous-like lesions, herpetiform lesions, candidiasis, and oral lesions of Kawasaki-like disease as the most common oral manifestations of COVID-19 disease [48]. Iranmanesh et al. (2020) have also reported tongue (38%), labial mucosa (26%), and palate (22%) as the most common sites of oral clinical course presentation. Similarly, the tongue 25% (n = 4) and buccal mucosa 25% (n = 4) were identified as the most common sites of presentation of the oral mucosal lesions in this prevalent cohort study.

In a case report, a 67-year-old Caucasian man who tested positive for coronavirus had presented with multiple oral manifestations such as recurrent herpes simplex, candidiasis, and geographic tongue [15]. Similarly, out of the total 16 patients who developed oral clinical courses, three were found to develop two symptoms each. Another systematic review and meta-analysis have also reported a 10%, 33%, and 44% prevalence rate of aphthous lesions, oral lesions, and xerostomia, respectively [49]. The findings of this cohort study also disclosed 62.5% (n = 10) and 31% (n = 5) prevalence rates of oral lesions and xerostomia, respectively. An additional systematic review that evaluated the association between oral health and COVID-19 reported the occurrence of hemorrhagic crust, bulla, buccal mucositis, and petechiae among patients with SARS-COV-2 infection [50].

Similarly, petechiae, buccal mucositis, hemorrhagic crust, and bulla were identified in this prospective cohort study. In general, as much of the oral clinical course presentations reported in several works of literature are obtained through interviewing patients [51], the findings of this prevalent cohort study could play a substantial role in filling the gap in observational clinical reality in the subject matter [42]. This crucial reality might explain the variation in various prevalence data [52]. Moreover, many of the oral afflictions seen in COVID-19 patients might not be directly caused by SARS-CoV-2 infection and, thus, should not be classified as oral manifestations of this disease [53]. Iatrogenic complications that occur in the course of the treatment of COVID-19 could be notable examples [54]. This group includes lesions caused by mechanical trauma of prolonged intubation and other invasive procedures employed [55].

The other factor that could potentially justify the variation in prevalence data is a drug; drug reactions might also present in the form of oral lesions [56]. Another group of oral lesions seen in patients with COVID19, but not directly related to the pathologic processes of SARS-CoV-2 infection, are opportunistic coinfections that involve the oral cavity [49]. In those regards, this prevalent cohort study has executed rigor inclusion criteria to avoid potential confounding. Of note, the necessity of conducting further in-depth observational studies needs to be underlined.

## Conclusion

The findings of this cohort study suggest that oral lesions are common manifestations in patients with COVID-19 infection. Comparatively, males were more prone to developing COVID-19-induced oral clinical courses. This indicates the necessity of conducting further observational studies to investigate the causal relationships between COVID-19 infection and the development of oral lesions as well as gender. As such, the findings of this study could be used as baseline data to conduct more comprehensive research with more study participants and a more extended follow-up period. On the other hand, the findings could also be used as scientific evidence to consider and initiate oral health care for patients with COVID-19. The findings could be crucial to advocating and optimizing patients' overall healthcare at the COVID-19 treatment centers. Therefore, this study suggests the necessity of multidisciplinary healthcare approaches to assure optimal health outcomes among patients diagnosed with COVID-19. Thus, the importance of the clinical dental examination of patients with infectious diseases in the OPD and ICU should be emphasized, considering the need for support, pain control, and quality of life. Additionally, the oral clinical courses of patients with COVID-19 could be used as potential early indicators of the SARS-CoV-2 infection.

## Supporting information

**S1 Fig. WHO oral assessment form for adults.**
(DOCX)

**S2 Fig. The modified oral assessment form for covid-19 patients.**
(DOCX)

## Acknowledgments

I want to express my most profound appreciation to Dr. Abdela Oumer for his valuable recommendations and cooperation during the making of this project.

## Author Contributions

**Conceptualization:** Heron Gezahegn Gebretsadik.

**Data curation:** Heron Gezahegn Gebretsadik.

**Formal analysis:** Heron Gezahegn Gebretsadik.

**Investigation:** Heron Gezahegn Gebretsadik.

**Methodology:** Heron Gezahegn Gebretsadik.

**Writing – original draft:** Heron Gezahegn Gebretsadik.

**Writing – review & editing:** Heron Gezahegn Gebretsadik.

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
