## [Decision Letter · Decision Letter 0]

12 Aug 2022

PONE-D-22-19013An update on oral clinical courses among patients with severe acute respiratory syndrome coronavirus 2 (SARS-CoV-2) infection: A clinical follow-up (a prospective prevalent cohort) studyPLOS ONE

Dear Dr. Gebretsadik,

Thank you for submitting your manuscript to PLOS ONE. After careful consideration, we feel that it has merit but does not fully meet PLOS ONE’s publication criteria as it currently stands. Therefore, we invite you to submit a revised version of the manuscript that addresses the points raised during the review process. Please clarify the inclusion criteria.

We look forward to receiving your revised manuscript.

Kind regards,

Etsuro Ito

Academic Editor

PLOS ONE

Journal Requirements:

2. We note you have included a table to which you do not refer in the text of your manuscript. Please ensure that you refer to Tables 1 and 2 in your text; if accepted, production will need this reference to link the reader to the Table.

Reviewers' comments:

Reviewer's Responses to Questions

**Comments to the Author**

1. Is the manuscript technically sound, and do the data support the conclusions?

Reviewer #1: Yes

2. Has the statistical analysis been performed appropriately and rigorously? 

Reviewer #1: No

3. Have the authors made all data underlying the findings in their manuscript fully available?

Reviewer #1: No

4. Is the manuscript presented in an intelligible fashion and written in standard English?

Reviewer #1: Yes

5. Review Comments to the Author

Reviewer #1: Line 55: As a recommendation, the authors can comment on the most recent influenza pandemic of 2009, which occurred 10 years before the new SARS-Cov-2 pandemic. In the introduction, the pandemic of more than 100 years ago (1918) is very well cited.

Line 83: At the end of the line, modify the citation (Xu et al., 2020), since the references are only numerical.

Line 107: Do not describe the objective as it is written in this section, because it is very well justified in line 112.

Line 127: It is essential to first propose the inclusion criteria. Identify the age ranges that can be included. There is a calculation to obtain the sample of the study population, it was random, it was stratified. It is a prospective study, it is very important to know how these parameters were made. If these methodologies were not considered in the project, it can be justified by writing that they included 55 patients on a certain date. As an example: from February 1, 2022 to March 1, 2022 (on the date that was the study protocol), 55 patients (36 men and 19 women) were included, in the format the age is asked, as a recommendation, they can calculate the ages of the men and women who participated. In the inclusion criteria, indicate what ages could be included in the study protocol.

Line 146: It is not mentioned which variables were collected and how they would be measured, it is only limited to an Excel sheet (an attached document on the study parameters is presented), the study variables must be specified and how they will be measured in the section of data analysis. This section must include the different variables that would be studied and how they will be measured, as an example the variables that are reported in table 1 of line 180 and in table 2 of line 197.

Line 148: The objective does not mention which variables will be collected and how they will be measured, place it in the section of line 146.

Line 162: The results are well written and clearly presented, but it would make it much easier to set the inclusion criteria, which variables would be reported and how these variables would be measured.

Line 286: The greatest strength of the article is the conclusion.

I consider that the paper can be accepted under the condition that all the proposed corrections of form and substance are made.

6. PLOS authors have the option to publish the peer review history of their article (what does this mean?). If published, this will include your full peer review and any attached files.

Reviewer #1: No

---

## [Author Response · Author response to Decision Letter 0]

6 Sep 2022

Dear Etsuro Ito,

I am delighted that my manuscript has been reviewed for publication by PLOS ONE. Thank you very much for your constructive feedback and helpful guidance.

I have taken the reviewer and editor’s comments very seriously and have invested a great amount of work to adequately address the concerns described in your decision email. I copied the reviewer and editor comments as presented in the email. For clarity, I bolded the comments as presented below. I responded to the editor’s and reviewer’s comments individually, indicating exactly how I addressed each concern or problem and describing the changes made. Two versions of the revised manuscript: one with highlights denoting where the text has been changed; the other a clean version are uploaded as the manuscript file. Contents removed and added to the highlighted manuscript are colored in red and green, respectively.

I hope that you approve the taken steps and look forward to hearing from you soon.

The author.

---

## [Decision Letter · Decision Letter 1]

26 Sep 2022

An update on oral clinical courses among patients with severe acute respiratory syndrome coronavirus 2 (SARS-CoV-2) infection: A clinical follow-up (a prospective prevalent cohort) study

PONE-D-22-19013R1

Dear Dr. Gebretsadik,

We’re pleased to inform you that your manuscript has been judged scientifically suitable for publication and will be formally accepted for publication once it meets all outstanding technical requirements.

Kind regards,

Etsuro Ito

Academic Editor

PLOS ONE

Reviewers' comments:

Reviewer's Responses to Questions

**Comments to the Author**

1. If the authors have adequately addressed your comments raised in a previous round of review and you feel that this manuscript is now acceptable for publication, you may indicate that here to bypass the “Comments to the Author” section, enter your conflict of interest statement in the “Confidential to Editor” section, and submit your "Accept" recommendation.

Reviewer #1: All comments have been addressed

2. Is the manuscript technically sound, and do the data support the conclusions?

Reviewer #1: Yes

3. Has the statistical analysis been performed appropriately and rigorously? 

Reviewer #1: Yes

4. Have the authors made all data underlying the findings in their manuscript fully available?

Reviewer #1: Yes

5. Is the manuscript presented in an intelligible fashion and written in standard English?

Reviewer #1: Yes

6. Review Comments to the Author

Reviewer #1: The requested changes were made to the article, I consider that the content is relevant, so the manuscript should be published.

7. PLOS authors have the option to publish the peer review history of their article (what does this mean?). If published, this will include your full peer review and any attached files.

Reviewer #1: No

---

## [Editor Report · Acceptance letter]

13 Oct 2022

PONE-D-22-19013R1 

An update on oral clinical courses among patients with severe acute respiratory syndrome coronavirus 2 (SARS-CoV-2) infection: a clinical follow-up (a prospective prevalent cohort) study 

Dear Dr. Gebretsadik:

I'm pleased to inform you that your manuscript has been deemed suitable for publication in PLOS ONE. Congratulations! Your manuscript is now with our production department. 

Kind regards, 

on behalf of

Prof. Etsuro Ito 

Academic Editor

PLOS ONE